# Decoupled few-femtosecond phase transitions in vanadium dioxide

Christian Brahms [1] ✉, Lin Zhang [2], Xiao Shen[3], Utso Bhattacharya [2,4], Maria Recasens [2], Johann Osmond[2], Tobias Grass [5,6], Ravindra W. Chhajlany[7], Kent A. Hallman[8], Richard F. Haglund [8], Sokrates T. Pantelides [8,9], Maciej Lewenstein [2,10], John C. Travers [1] & Allan S. Johnson [11] ✉

The nature of the insulator-to-metal phase transition in vanadium dioxide (VO$_2$) is one of the longest-standing problems in condensed-matter physics. Ultrafast spectroscopy has long promised to determine whether the transition is primarily driven by the electronic or structural degree of freedom, but measurements to date have been stymied by their sensitivity to only one of these components and/or their limited temporal resolution. Here we use ultra-broadband few-femtosecond pump-probe spectroscopy to resolve the electronic and structural phase transitions in VO$_2$ at their fundamental time scales. Our experiments show that the system transforms into a bad-metallic phase within 10 fs after photoexcitation, but requires another 100 fs to complete the transition, during which we observe electronic oscillations and a partial re-opening of the bandgap, signalling a transient semi-metallic state. Comparisons with tensor-network simulations and density-functional theory calculations show these features result from an unexpectedly fast structural transition, in which the vanadium dimers separate and untwist with two different timescales. Our results resolve the structural and electronic nature of the light-induced phase transition in VO$_2$ and establish ultra-broadband few-femtosecond spectroscopy as a powerful tool for studying quantum materials out of equilibrium.

Strong correlations between internal degrees of freedom create the extraordinary properties of quantum materials, but also make it challenging to identify which interactions underlie any specific phenomenon. The insulator-to-metal transition (IMT) in vanadium dioxide (VO$_2$) exemplifies this challenge[1–3]: whether the IMT is primarily electronically or structurally driven has been debated for over 50 years. In the low-temperature monoclinic insulating (M1) phase of VO$_2$, the vanadium ions form pairs of twisted dimers (Fig. 1a). The vanadium 3$d$-orbitals form bands of $d_\parallel$ and $\pi^*$ symmetry (Fig. 1b) oriented along and away from the dimer chains, respectively (Fig. 1c). The $d_\parallel$ band splits into a filled bonding and an empty anti-bonding band, while the $\pi^*$ bands lie above the Fermi energy, opening a bandgap. Upon heating above 340 K, both the band splitting and structural distortion disappear and VO$_2$ transitions to a metallic rutile (R) structure with evenly

[1]School of Engineering and Physical Sciences, Heriot-Watt University, Edinburgh, UK. [2]ICFO - Institut de Ciencies Fotoniques, The Barcelona Institute of Science and Technology, Castelldefels (Barcelona), Spain. [3]Department of Physics and Materials Science, University of Memphis, Memphis, TN, USA. [4]Institute for Theoretical Physics, ETH Zurich, Zurich, Switzerland. [5]DIPC - Donostia International Physics Center, San Sebastian, Spain. [6]IKERBASQUE, Basque Foundation for Science, Bilbao, Spain. [7]Institute of Spintronics and Quantum Information, Faculty of Physics and Astronomy, Adam Mickiewicz University, Poznań, Poland. [8]Department of Physics and Astronomy, Vanderbilt University, Nashville, TN, USA. [9]Department of Electrical and Computer Engineering, Vanderbilt University, Nashville, TN, USA. [10]ICREA, Barcelona, Spain. [11]IMDEA Nanoscience, Madrid, Spain. ✉e-mail: c.brahms@hw.ac.uk; allan.johnson@imdea.org

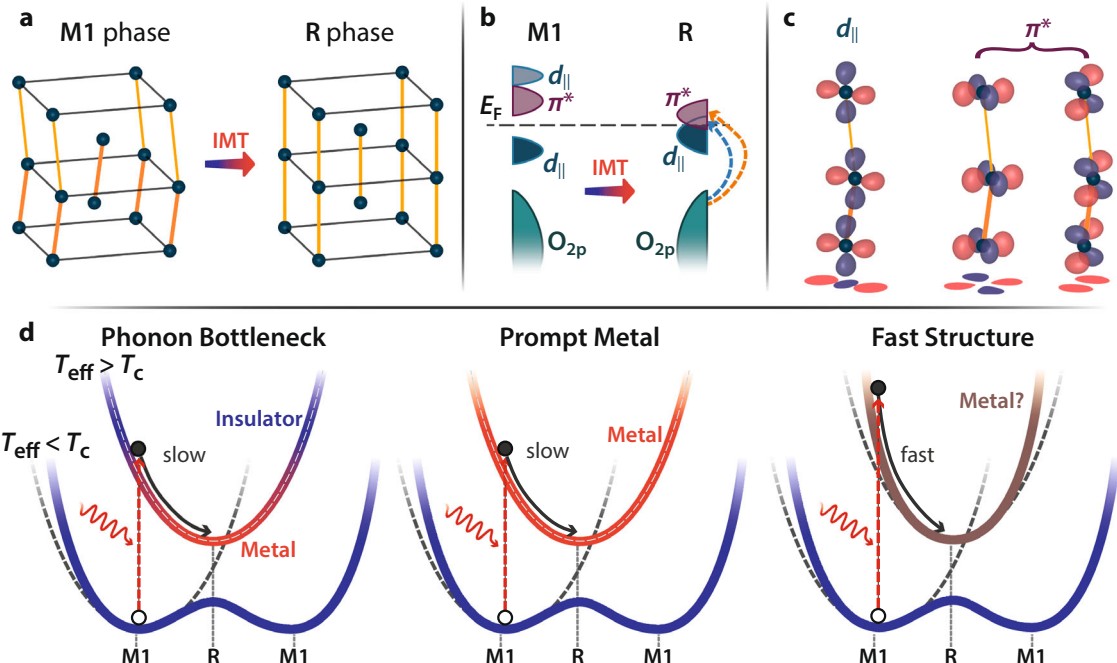

**Fig. 1 | The light-induced phase transition in VO₂. a** Structure of VO₂ in the monoclinic (M1) and rutile (R) phases. Oxygen atoms are omitted for clarity. Darker orange lines indicate shorter (dimerised) bonds in the M1 phase. **b** Associated electronic density-of-states changes between the phases. In the M1 phase, the $d_{\parallel}$ band splits and the $\pi^*$ orbitals move to higher energies, opening a bandgap. Also indicated are the main optical transitions in the R phase as determined by ellipsometry (see methods). **c** Orbital structure of the two main bands involved in the IMT. Red and blue colouring indicate positive and negative values, respectively. The dimerisation and tilt in the M1 vanadium chains change the overlap of the $d_{\parallel}$ orbitals and two $\pi^*$ orbitals. **d** Possible scenarios for the interplay of the structural

and electronic components of the ultrafast IMT. Photoexcitation (red arrows) causes a sudden change of lattice potential and raises the effective temperature $T_{eff}$ above its critical value for the phase transition, $T_c$. In the phonon bottleneck picture (left), VO₂ remains insulating (blue shading) until the structure transforms (grey arrow), while in the prompt metal scenario (centre), the system abruptly metallises (red shading) and the structure transforms later. In the fast structural scenario (right), the lattice moves faster than the linear phonon modes, leaving it unclear if the structural transformation follows the electronic one or vice versa (brown shading). Dashed grey lines show the harmonic approximation to the lattice potential.

spaced vanadium ions and a single $d_{\parallel}$ band at the Fermi level (Fig. 1a, b). However, whether a Peierls-like structural instability and crystal-field splitting of the $d_{\parallel}$ band is sufficient to produce the bandgap has long been debated[3]. In another interpretation, VO₂ is a Mott insulator, so the $d_{\parallel}$ band is split primarily by electron correlations, and the structural transition is secondary[1].

Ultrafast spectroscopy has been a key tool used to address this and similar questions, because it can decouple the normally concurrent changes in different degrees of freedom[4]. Intuitively, in a structurally driven IMT, the electronic system cannot metallise until the slow structural transition has occurred. An 80 fs speed limit in the electronic response in early experiments matched the period of the phonon modes connecting the M1 and R phases and was taken as evidence of a structurally driven transition[5] (Fig. 1d, left). However, later time-resolved photoelectron experiments then showed the metallization actually occurs in less than 60 fs (resolution limited)[6], while recent experiments using attosecond extreme-ultraviolet and/or few-cycle infrared pulses observed transition times as low as 25 fs[7,8], far below the phonon period. These observations suggested a Mott-like transition, in which VO₂ metallises instantaneously while remaining structurally monoclinic and completes the transition to rutile later (Fig. 1d, middle). Because these experiments were predominantly sensitive to the electronic degree of freedom, their interpretations share the a priori assumption that the linear phonon modes of the M1 phase limit the speed of lattice motion[5,7,8]. However, the vanadium dimers displace by $\approx 0.3$Å during the phase transition, well into the nonlinear phonon limit[9]. This could lead to rapid structural motion with or without simultaneous metallisation (Fig. 1d, right). Ultrafast diffraction measurements show that the structure transforms in 100 fs

or less[10–12], but this is limited by the experimental time resolution. In all cases, these electronic or structurally sensitive measurements observed a simple monotonic switching in which the assignment of structural or electronic driven nature was made based on heuristic timescale arguments[7,8,11,13–15]. Therefore, fully disentangling the two degrees of freedom requires measurements with high time resolution and simultaneous structural and electronic sensitivity.

Here, we study the photoinduced IMT in VO₂ via ultrafast optical spectroscopy with ~5 fs resolution and a probe spectrum spanning from the deep ultraviolet to the infrared, which we generate via soliton self-compression and resonant dispersive wave emission in gas-filled hollow capillary fibres[16,17]. The ultra-broadband spectral window allows us to probe all valence and conduction band transitions simultaneously and observe the collapse of the bandgap, rise of the metallic Drude response, and evolution of the charge-transfer bands[18]. Tracking the full dielectric response guarantees sensitivity to structural changes, as the 1st-order coupling of light is not to free electrons but to their dipole with the nuclei. From this, we can build a complete picture of the changes in both the electronic and structural degrees of freedom with femtosecond time resolution. In sharp contrast to previous measurements, we find that the system follows a complex pathway. Within 10 fs after photoexcitation, the system is best described as a very bad metal[18,19] with a Drude scattering time of only 0.1 fs—to the best of our knowledge, the shortest ever observed. Surprisingly, this highly non-equilibrium state persists for over 100 fs despite this scattering, suggesting an electronic bottleneck scenario. In parallel, the $d_{\parallel}$ bands at the Fermi edge exhibit unexpectedly complex, 20 fs-scale changes which also decay over 100 fs. These results suggest that both the electronic configuration and the structure evolve very quickly after

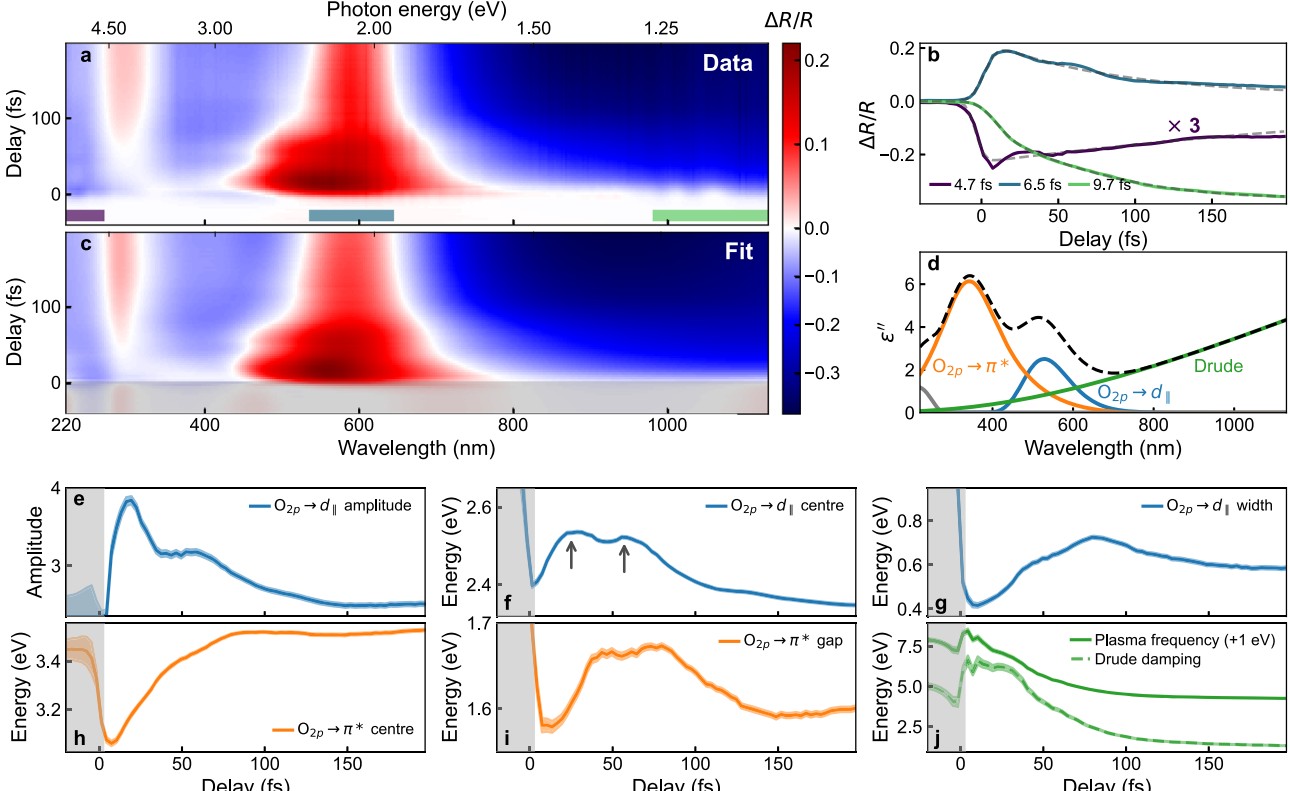

**Fig. 2 | Ultra-broadband ultrafast spectroscopy of the photoinduced phase transition in VO₂. a** Transient reflectivity of a 45 nm thick VO₂ thin film on sapphire when excited through the phase transition by a few-fs-duration pulse centred at 610 nm and a fluence of 29 mJ/cm². A variety of non-trivial dynamics are seen at all regions of the spectrum. **b** Lineouts (coloured lines) and rise time fits (dashed lines) of the dynamics averaged over wavelength in the three spectral bands indicated in (**a**). Transition times as low as 5 fs (temporal resolution limited) are observed. **c** Fit of the transient differential reflectivity using the metallic phase dielectric function. **d** The dielectric function of the rutile metallic phase (dashed line), as well as the constituent resonances (solid). Only the Drude term and the two indicated optical

transitions (coloured lines) are allowed to vary in fitting the experimental data in panel **a**. The $O_{2p} \rightarrow d_\parallel$ resonance is modelled as a Gaussian while the $O_{2p} \rightarrow \pi^*$ is modelled as a Tauc-Lorentz resonator. **e**–**j** Variation of the fitting parameters from (**d**) with pump-probe delay. All parameters of the Drude and $O_{2p} \rightarrow d_\parallel$ transitions are allowed to vary freely, while for the $O_{2p} \rightarrow \pi^*$ resonance the width and amplitude are fixed. This set of parameters represents the minimal best fit to the data; further details on the fitting procedure and model in Methods. The grey shaded areas around the lines show the error on the fit parameters as given by the square root of the diagonal of the covariance matrix.

photoexcitation, with no speed limit imposed by the linear phonon modes. Our interpretation is supported by advanced electronic tensor network with self-consistent phononic mean field (TN-MF) calculations of a dimer chain, which show that the two structural distortions of the monoclinic phase disappear on different sub-50 fs timescales. Additional density-functional theory (DFT) calculations confirm that such a transient configuration is metallic and exhibits density-of-states bottlenecks which slow down hot-electron relaxation.

## Results

In our experiments, we induce the IMT in VO₂ with few-femtosecond pump pulses centred around 610 nm and probe the delay-dependent reflectivity of our sample with a sub-cycle-duration supercontinuum spanning 220 nm to 2550 nm[17] (Methods and Supplementary Fig. 1). Figure 2a shows the experimentally measured delay-dependent differential reflectivity ($\Delta R/R$) of our 45 nm thick VO₂ film between 220 nm and 1130 nm at a pump fluence of 29 mJ/cm², sufficiently above the phase transition threshold of 10 mJ/cm² to ensure full transformation of the film (Supplementary Fig. 2). Three key features are evident: i) at temporal overlap (zero delay), the reflectivity changes very abruptly across the entire spectral window; ii) the subsequent evolution is complex with rapid changes; and iii) this initial evolution is completed within around 100 fs. This complex behaviour is in sharp contrast to previous studies of VO₂[5–8,13,20], and is revealed only via our improved temporal resolution and spectral bandwidth. To quantify the

speed of the initial response, Fig. 2b shows kinetic traces in three wavelength bands indicated in Fig. 2a, along with fits to an initial step convolved with a double exponential decay (dashed lines). In all three bands, the response time is below 10 fs; in the ultraviolet it is below 5 fs. This response time is the fastest yet recorded in VO₂[7,8] and suggests that this part of the transition is effectively instantaneous.

For a more detailed analysis, we leverage our ultra-broadband probe and extract the evolution of the Drude plasma and vanadium 3d bands. Using auxiliary static measurements, we convert $\Delta R/R$ at each time delay into an absolute reflectivity R. We then fit $R(\omega)$, where $\omega$ is photon energy, using the metallic rutile-phase permittivity, $\varepsilon(\omega) = \varepsilon'(\omega) + i\varepsilon''(\omega)$. We follow the model of Ref. 21, in which the relevant transitions are those from the filled $O_{2p}$ band to the empty vanadium $d_\parallel$, $\pi^*$ and $\sigma^*$ bands, as well as the intraband Drude plasma (Methods). Figure 2d shows the resulting $\varepsilon''$ at 200 fs delay (black dashed) and relevant constituent transitions (coloured lines). Detailed lineouts comparing the fit to the experimental data are shown in Supplementary Fig. 3. Both the total response and the underlying components agree well with previous results[18,21,22]. We can accurately describe the dynamics allowing only the $d_\parallel$ and $\pi^*$ bands and Drude term to vary (Methods), in good agreement with the general understanding of the IMT[2,23]. Figure 2c shows $\Delta R/R$ for the resulting fit and Fig. 2e–j the time evolution of the fit parameters. The grey shaded areas indicate negative delays, for which the sample is in the M1 phase. As the R-phase dielectric function cannot describe this phase, the

parameters in this region are spurious. For positive delays, the fit captures the data remarkably well, indicating that the sample metallises immediately upon photoexcitation. This complete sensitivity to the electronic structure allows us to go beyond previous high-time resolution experiments which primarily reported on charge screening effects[7,13]. After 100 fs, the optical response closely resembles that of the thermal rutile phase and the evolution slows down significantly, in excellent agreement with the fastest measurements with direct structural sensitivity[10,11].

We first focus on the Drude response (Fig. 2j). The free-carrier density is initially around 50% higher than in the thermal metallic phase and decays monotonically. The damping coefficient immediately after photoexcitation is 6.6 eV; as shown in Supplementary Fig. 4, our data cannot be modelled without such a high damping coefficient. This corresponds to an electron scattering time of only $\sim 0.1$ fs and indicates very strong electron-electron interactions−remarkably strong even for $VO_2$, which in equilibrium is a bad metal with scattering times below the Mott-Ioffe-Regel limit[18,24]. Such rapid scattering should lead to very fast relaxation. The $\sim$ roughly 100 fs timescale for relaxation, closer to that of conventional metals with lower scattering rates, thus implies a barrier to electron-hole recombination. A comparison can be drawn to the case of photo-excited graphene where, despite high electron-electron scattering rates, a DOS bottleneck at the Dirac point leads to slow phonon-mediated relaxation[25]. As we will show later, a similar DOS bottleneck occurs in photoexcited $VO_2$.

We next consider the dynamics of the vanadium $3d$ bands (Fig. 2e–j). Multiphoton absorption is known to be very weak in the M1 phase[26] and the photon energy of our pump pulse is insufficient to directly excite holes in the $O_{2p}$ orbitals. The $O_{2p}$ orbitals themselves exhibit only marginal changes during the IMT[23]. Therefore, the $O_{2p} \rightarrow d_\parallel$ and $O_{2p} \rightarrow \pi^*$ transitions effectively track the motion of the unoccupied portion of the $d_\parallel$ and $\pi^*$ bands above the Fermi level. The $O_{2p} \rightarrow d_\parallel$ and $O_{2p} \rightarrow \pi^*$ transitions are fit by a Gaussian and Tauc-Lorentz respectively[21]. There are clear double-peak revival structures on a 20 fs timescale, most obvious in the amplitude and central energy of the $d_\parallel$ transition (Fig. 2e, f). Similar structures can be seen in Fig. 2g, i. This motion cannot result from coherent electronic effects as it is much slower than the scattering times for conduction-band electrons. Conversely, it is faster than the phonon modes previously considered to govern the phase transition[5,27,28]. As shown in Fig. 2h, the $\pi^*$ band energy moves monotonically over a similar timescale to the Drude response, while the dynamics of the gap term shown in Fig. 2i agree qualitatively with those of the $d_\parallel$ band. This shows internal consistency of the fits, as the gap term in a Tauc-Lorentz model is related to the next-lowest-lying band[29].

The motion of the $d_\parallel$ and $\pi^*$ transitions leads us to a surprising conclusion even before considering its origin. The $O_{2p} \rightarrow d_\parallel$ transition promotes electrons to just above the Fermi level (see Fig. 1b). After less than 10 fs, it has a central energy of 2.4 eV and bandwidth of 0.4 eV (Fig. 2f, g), very close to equilibrium metal values (2.35 eV and 0.6 eV, respectively). This shows that the $d_\parallel$-band splitting collapses quasi-instantaneously, in good agreement with the rapid metallisation. However, after 30 fs the $d_\parallel$ transition has moved upward to 2.55 eV with 0.45 eV bandwidth (Fig. 2f, g). This shift and broadening indicate a reduced density of states (DOS) at the Fermi level (see Supplementary Fig. 5). In particular, the central-energy change closely matches the 0.2 eV shift of the unoccupied bands during the thermal IMT[23]. Our data thus suggests that within 30 fs, the direct band gap partially re-opens and $VO_2$ becomes transiently semimetallic. This explains the slow relaxation of the Drude plasma: while intraband collisions are very frequent, a transient DOS bottleneck near the Fermi level limits electron-hole recombination until the $d_\parallel$ states have completely relaxed, which takes around 100 fs[25]. We note the reduced width of the $d_\parallel$ band as compared to the thermal R-phase is also a plausible explanation for the increased intraband scattering rate.

At this point we can see that $VO_2$ transforms to a bad-metallic state in less than 10 fs and that the electronic structure then undergoes a complex evolution resulting in a transient semi-metallic state. The relatively slow oscillatory motion of the bands naturally suggests a structural origin for this post-10 fs evolution, but the timescale is still well below the characteristic phonon modes for the vanadium dimers. We briefly note that a structural origin for these dynamics would remain consistent with the most recent diffraction data[10,11,30], which show a mixture of coherent and disordered structural dynamics below 100 fs, though these works lack the temporal resolution to report on the dynamics seen here. To understand the dynamics of the $d_\parallel$ and $\pi^*$ bands and their likely structural origin, we have performed calculations of the light-induced phase transition using a TN-MF approach[31]. Motivated by the dominance of the $d_\parallel$ and $\pi^*$ bands, we model a two-band system and solve the fully quantum equations of motion for the electronic degree of freedom. We treat the nuclei classically and model the dimerisation and tilt as two independent components in a sixth-order Landau potential to describe the $1^{st}$-order nature of the phase transition[3]. We validate this approach by recovering the thermal phase transition using finite-temperature tensor-network calculations and describe photoexcitation with a Peierls substitution; further details in the methods section and ref. 31.

Figure 3 shows the results of these simulations. In agreement with experiment, we find a prompt collapse of the bandgap in the optical conductivity after only 13 fs (Fig. 3a). More surprisingly, the two components of the structural distortion (Fig. 3b) transform in just tens of femtoseconds (Fig. 3c), faster than the linear phonon modes of the system and in agreement with the timescales for motion of the $d_\parallel$ and $\pi^*$ transitions observed in experiment. In the simulations, the displacement of the dimers ($X_1$) is lost faster than the tilt ($X_2$)−similar to early proposals about the structural transformation in $VO_2$[27], but several orders of magnitude faster. The high velocity results in significant inertia, whereby the displacement undergoes a transient revival with opposite sign[15]. Similar effects have been predicted in recent time-dependent DFT calculations[32], though here we find no strong dependence on the photocarrier doping level, in agreement with recent diffraction measurements[30].

These findings are in excellent agreement with our experimental data and provide a plausible explanation for the complex evolution of the electronic state. The $d_\parallel$ band is strongly sensitive to both displacement and tilt of the dimers as they affect the overlap of the orbitals along the dimerization direction. The oscillations in the $d_\parallel$ band can then be explained as due to (a) the loss and subsequent partial restoration of the displacement leading to an oscillation of the orbital overlap and (b) the separate timescale for the evolution of the tilt. Conversely, the $\pi^*$ orbitals are aligned perpendicular to the dimer chains and hence less sensitive to the displacement. This band is governed either by tilt, which disappears more slowly and monotonically, or by the relaxation of carriers.

To further verify that the band dynamics can be explained by structural changes, we have compared our results to electronic structures calculated using DFT based on a hybrid exchange functional (see Methods). In particular, the TN-MF dynamical calculations predict that the system passes through a metallic phase with greatly reduced dimerization and tilt as compared to the M1 phase. This state is similar to the recently proposed M0 phase[33]. In Fig. 3d, e we compare the DOS calculated for the M1, M0 and R phases using this approach (Methods). A single hybrid exchange functional is used to describe all three phases, making it uniquely suited to examine out-of-equilibrium states. Compared to the R phase, the M0-phase DOS shows several minima near the Fermi level (−0.4 eV to 0.6 eV) and changes from predominantly $\pi^*$ character to predominantly $d_\parallel$ in this region. As our 2 eV pump pulse primarily excites $d_\parallel \rightarrow \pi^*$ transitions in the M1 phase[21], the photoexcited carriers are initially centred at an energy of 1.5 eV, as indicated by the red Gaussian in Fig. 3d. This feature supports our

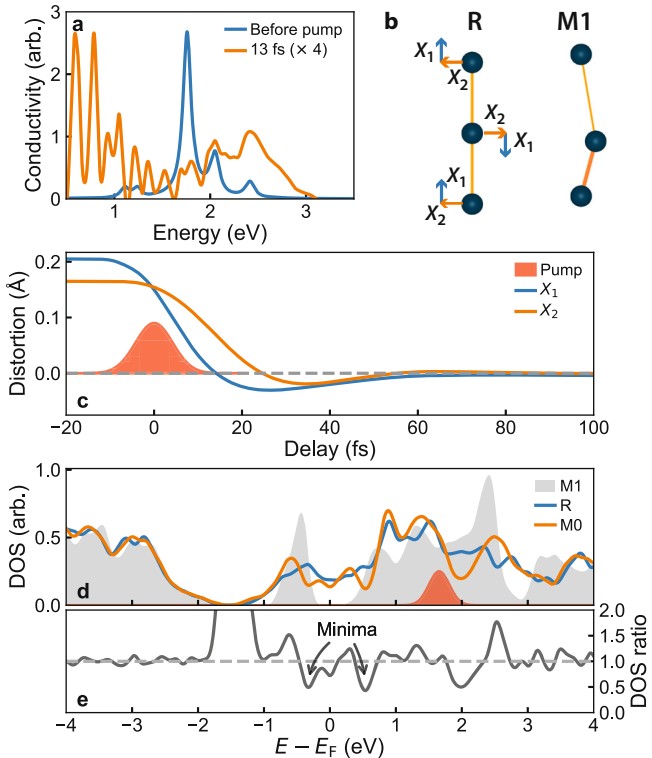

**Fig. 3 | Simulations of the ultrafast phase transition. a** TN-MF simulated optical conductivity before and 13 fs after photoexcitation showing instantaneous band-gap collapse. **b** Illustration of the two components of the structural distortion. **c** Structural dynamics following the phase transition. The dimerization relaxes prior to the tilt, overshooting the rutile positions, before fully relaxing in less than 100 fs. **d** DOS calculated with DFT for the monoclinic and rutile phases, and for a structural configuration (M0) close to that predicted by the TN-MF calculations for ~20 fs delay. **e** Ratio of the DOS of the M0 configuration and the rutile phase. The DOS near the Fermi level in the M0 phase exhibits several minima as compared to the rutile phase. Electrons generated at energies excited by the pump (red Gaussian in **d**) need to relax across these minima prior to full thermalization.

interpretation of the experimental data: the hot carriers have to relax across DOS bottlenecks, so the non-equilibrium state can persist for much longer than would be expected.

## Discussion

We can now provide a comprehensive description of the light-induced IMT in $VO_2$ at the shortest timescales, as schematically illustrated in Fig. 4. Photoexcitation causes a purely electronic collapse of the bandgap and transition to a monoclinic metallic state within 10 fs. This metallic state is characterised at first by remarkably bad metallicity[18,24], and then by semi-metallicity caused by the competing crystal-field splitting that favours re-opening the bandgap[2]. The fast electronic rearrangement creates a large, non-adiabatic force on the vanadium ions. In response, the ions move towards their rutile positions faster than the characteristic phonon frequencies of the monoclinic phase. As the structure relaxes, the crystal-field splitting of the conduction band reduces, but structural overshoots due to the large ion inertia can transiently revive this feature. After just 100 fs, the structure fully relaxes due to the strong electron-lattice coupling[10] and the system approaches the thermal metallic state, with strain- and temperature-propagation effects occurring on longer timescales[14]. Previous measurements have reported only on the quasi-adiabatic post-50 fs dynamics[6,10,11,30], by which time the emergent bad-metallic nature, semi-metallic transient state, and non-linear phonon dynamics have all decayed and the state is simply characterized by much more conventional heating and strain.

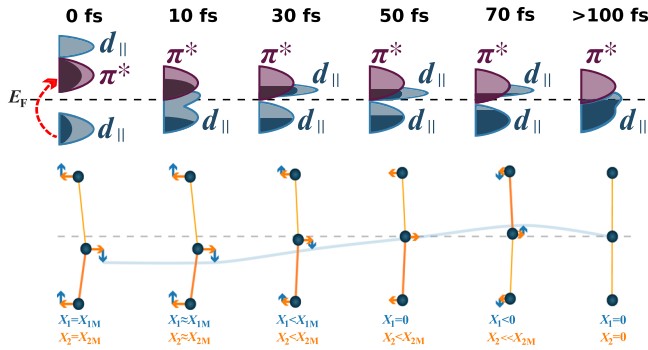

**Fig. 4 | Electronic and structural changes during the light-induced IMT of $VO_2$.** The system begins in the M1 phase, then photoexcitation causes a prompt electronic phase transition and the bandgap collapses. This launches rapid and coherent phonon motion, causing the lattice to transform faster than would be expected from the normal phonon modes. The structural competition/electron cooling causes the electronic structure to partially re-open the bandgap as the dimerization and tilt coherently evolve, stymying further purely electronic relaxation. After 100 fs strong electron-phonon scatters cools the electrons and heats the lattice, damping all coherent motion, and the system approaches the thermal rutile phase.

Our results reveal the surprising complexity of the early-time dynamics of the light-induced phase transition in $VO_2$. In equilibrium, a cooperative effect between Mott-like electronic interactions and crystal-field splitting has been suggested to explain the bandgap in the monoclinic phase[3,34]. Here, we find that these effects can be competitive when out of equilibrium, and that the quasi-instantaneous electronic re-arrangement can trigger non-linear lattice dynamics. Because the electronic structure transforms prior to structural rearrangement, the system transiently passes through a monoclinic metallic state. This suggests it should be possible to drive an electronic-only phase transition, in qualitative agreement with a Mott picture[7,8]. However, our results also indicate that any such phase would be very short-lived: the strong electron-phonon coupling would transfer energy to the lattice bath in less than 100 fs[10], so the system would revert to the insulating state unless sufficient energy is provided to also trigger the structural transformation. Any monoclinic-phase metal that does not subsequently transform to the rutile phase would, therefore, likely be found in a narrow fluence range just below the transition threshold and have a lifetime of at most a few hundred femtoseconds. Our results also demonstrate the power of ultra-broadband, few-femtosecond spectroscopy enabled by soliton self-compression and resonant dispersive wave emission[16,17] for understanding dynamics in the condensed phase. By directly tracking all relevant optical transitions for band-structure changes, we can build a comprehensive picture of the phase transition at short time scales which are inaccessible with other techniques. We note that the light-induced phase transition in $VO_2$ is one of the most heavily studied of all phenomena in quantum materials; despite this, our methodology was able to uncover a surprising and unexpectedly complex pathway. This suggests that similarly complex pathways could exist in practically any other material characterized by strong couplings, and that novel non-linear optical effects and control pathways for light-induced phase transitions could exist at the sub-50 fs timescale across a broad class of materials. For instance, modifying the photoexcitation energy alone, generally taken to have a minimal impact on the asymptotic time behaviour in many materials[35], could drastically alter these initial dynamics. Our methodology will thus find future applications in understanding other light-driven quantum materials[36] where broad resonances and multiple energy scales can challenge conventional methodologies, as well as enabling optical attosecond spectroscopy in the condensed phase[17].

## Methods

### Sample deposition and characterization

An approximately 45 nm thick thin film of $VO_2$ was deposited onto a 001 (c-cut) sapphire substrate using pulsed laser deposition at room temperature and subsequent annealing[37]. The mean-square strain was measured using the Williamson-Hall method to be 0.13, consistent with an earlier determination for a similar film. The dielectric properties and thickness were determined using ellipsometry. Data were taken at three angles of incidence (60, 65, and 70 degrees) and wavelengths spanning from 200 to 2700 nm, which were then fit with a similar model to that proposed in Qazilbash et al. [21], yielding good agreement with the parameters determined there. A film thickness of 45 nm and a penetration depth of 65 nm at 660 nm wavelength were determined. Upon heating above 340 K a sharp change in the optical properties was observed as the sample underwent the phase transition into the rutile metallic phase. The full reflectivity of the rutile metallic phase was then fit to the same dielectric functions as in Qazilbash et al. [21], with excellent agreement found in the dielectric parameters as compared to previous literature. In particular the imaginary part of the dielectric function was modelled as:

$$\varepsilon'' = \frac{A_{UV}}{\omega_{UV} - \omega} + \frac{A_{IR}}{\omega}$$
$$+ \text{Im}\left\{ G\left(\omega, A_{d_{\parallel}}, \omega_{d_{\parallel}}, \sigma_{d_{\parallel}}\right) + TL\left(\omega, A_{\pi^*}, \omega_{\pi^*}, \sigma_{\pi^*}, g_{\pi^*}\right) + TL\left(\omega, A_{\sigma^*}, \omega_{\sigma^*}, \sigma_{\sigma^*}, g_{\sigma^*}\right) - \frac{\omega_p^2}{\omega^2 + i\gamma\omega} \right\}$$

$$(1)$$

where $\omega$ is the photon energy, G denotes a causality-corrected Gaussian function[38] and TL denotes a Tauc-Lorentz function[29]; the real part is related via the Kramers-Kronig relations. The terms in the expression are a UV pole, IR pole, Gaussian resonance describing the $O_{2p} \rightarrow d_{\parallel}$ transition, a Tauc-Lorentz describing the $O_{2p} \rightarrow \pi^*$ transition, a Tauc-Lorentz describing the $O_{2p} \rightarrow \sigma^*$ transition, and the Drude plasma term. The Drude plasma term is determined by the plasma frequency $\omega_p$ and damping term $\gamma$. Apart from the Drude term, in each case $A_X$, $\omega_X$, and $\sigma_X$ denote the amplitude, resonant frequency and width of resonance X. The Tauc-Lorentz resonances include an additional term $g_X$ which denotes the gap term. In a semiconductor or insulator the gap term denotes the energy of the bandgap; for photon energies below this, the probability of the optical transition goes to zero, leading to a strong asymmetry in the resonance structure. In materials without a bandgap there is no sharp turn-on energy to the resonance, and this term instead relates approximately to the next lowest lying energy band into which carriers can be promoted instead of to the Tauc-Lorentz resonance, leading to a more gradual turn-on of the resonance.

### Ultrafast spectroscopy

A schematic of the experimental apparatus for ultra-broadband few-femtosecond spectroscopy is shown Supplementary Fig. 6. A titanium-doped sapphire amplifier (Coherent Legend Elite Duo USP) delivers 35 fs pulses (full width at half-maximum, FWHM) with 8 mJ of energy at 1 kHz repetition rate. These are converted to 1.45 mJ pulses at 1750 nm in an optical parametric amplifier (Light Conversion TOPAS-PRIME-HE). The infrared pulses are spectrally broadened in a gas-filled hollow capillary fibre (HCF) with 700 µm core diameter and 1.25 m length which is filled with argon gas at 450 mbar pressure, and subsequently compressed by propagation through bulk material. This results in 12 fs FWHM pulses with around 1 mJ of energy. A beamsplitter divides the beam into two, and both beams pass through variable attenuators formed of an achromatic half-wave plate and a silicon plate at Brewster's angle before being coupled into separate HCFs for the generation of pump and probe pulses.

The pump pulse is generated via soliton self-compression and resonant dispersive wave (RDW) emission in a 1.3 m HCF with 450 µm core diameter filled with 1050 mbar of argon. When driving this HCF

with around 135 µJ of pulse energy (measured before the HCF entrance window), around 22 µJ of energy is converted into an RDW pulse centred at 610 nm with a transform-limited pulse duration of approximately 3.5 fs. After exiting the HCF system this pulse passes through a 0.5 mm $CaF_2$ beam sampler and a motorised chopper wheel and then into the vacuum chamber. After collimation with a spherical metallic mirror, four chirped mirrors (Ultrafast Innovations PC70) both compensate for the dispersion of optical components and act as dichroic mirrors to remove the infrared supercontinuum from the pump beam. An in-vacuum half-wave plate and silicon-plate form an attenuator for the pump pulse. The filtered and attenuated pulse is then focused onto the $VO_2$ sample with a spherical metallic mirror to a spot size of approximately 200 µm FWHM. Supplementary Fig. 1b shows the result of an in-situ self-diffraction cross-correlation frequency-resolved optical gating (SD-XFROG) measurement of the pump pulse, taken by replacing the $VO_2$ sample with a 170 µm thick piece of silica glass and one of the wedges in the probe arm with a mirror and measuring the transmitted pump spectrum as a function of delay. While the pump pulse profile cannot be retrieved from this measurement without precise knowledge of the on-target probe spectrum and/or pulse profile, the trace clearly shows a single isolated pulse with negligible pre- or post-pulse structure (note the logarithmic colour scale). The temporal width of the cross-correlation, obtained by integrating the SD-XFROG trace over all wavelengths, is consistent with a root-mean-square time resolution of ∼ 5 fs, which in turn is consistent with the fastest feature observed in Fig. 2b.

A second HCF with 200 µm core diameter and 40 cm length provides the probe pulses. This HCF is directly connected to the vacuum chamber, creating a pressure gradient; the argon pressure in the entrance cell is maintained at 800 mbar with an electronic flow regulator. With 150 µJ of driving energy (measured before the HCF entrance window), a supercontinuum spanning 220 nm to 2550 nm is generated by soliton self-compression. After collimation with a spherical mirror with a UV-enhanced aluminium coating, the probe beam is attenuated by reflection from two wedged glass windows before being refocused to a spot size of approximately 38 µm FWHM on the sample. The reflected probe beam is then reimaged with 6x magnification onto the 25 µm entrance slit of a CCD spectrometer (Avantes ULS2048XL, detection range 200–1160 nm with 1715 pixels) outside the vacuum chamber by another spherical mirror, thus sampling only the central region of the probe. A half-wave plate before the probe generation HCF rotates the probe polarisation to be opposite to that of the pump; a polariser just before the spectrometer thus passes the probe but suppresses scatter from the pump.

Pump-probe measurements are acquired in a single-shot pump-on/pump-off scheme. The chopper wheel is synchronised to switch the pump pulse at half the laser repetition rate. The pump-probe delay is scanned in steps of 1 fs by a motorised delay stage in the probe arm. 2000 probe shots are acquired at each delay (with the pump on for half of these).

We calculate the differential reflectivity as $\Delta R/R = (I_{\text{on}} - I_{\text{off}})/I_{\text{off}}$, where $I_{\text{on}}$ and $I_{\text{off}}$ are the spectra with the pump on and off, respectively. To reduce the effect of detector noise, we average all spectra in five-pixel bins along the wavelength axis, resulting in wavelength samples separated by approximately 3 nm. Because of minor alignment drifts between pump-probe scans, we numerically correct the delay axis of each scan by overlapping the fastest delay-dependent feature in our data (see Supplementary Fig. 7). To remove fast oscillations caused by interference with residual pump scatter, we then re-bin along the delay axis, resulting in a spacing of 3 fs between delay points. Beyond removing the fast oscillations this does not affect the observed dynamics (see Supplementary Fig. 8). We then further denoise the data through a principal-component analysis (see Supplementary Fig. 9). We retain the four most significant components, which captures all the salient features in our experimental data.

## Dynamics Fitting

The differential reflectivity reported in Fig. 2a is first converted into an absolute reflectivity using the reflectivity of the M1 phase extracted from ellipsometry. The reflectivity at each time step as a function of probe photon energy is then fit using the dielectric function of the rutile metallic phase; we begin by fitting the long time delays (200 fs) at which the reflectivity agrees well with the thermal metallic state, and then move backwards in pump-probe delay, updating our initial guess with the output from the previous time delay. The full dielectric function has 14 free parameters and clearly overfits the data. Instead, we systematically fix a number of parameters to agree with the rutile metallic phase values and then allow the remaining parameters to vary freely. We examine all permutations of this fitting procedure (1969 combinations). The seven parameters shown in Fig. 2c are found to be the optimal minimal set which can accurately describe the phase transition for the following reasons: (1) they are the set that minimizes the error when allowing seven parameters to vary. (2) When allowing fewer parameters to vary, the optimal parameters are always from this set of seven parameters. (3) They show self-consistency, particularly in the dynamics of the Tauc-Lorentz gap term (Fig. 2i) and the Gaussian transition (Fig. 2f, g), in which the gap term should be expected to track the edge of the next lowest transition. (4) Adding additional parameters induces unphysical behaviour in the time-behaviour, namely sudden jumps. (5) Qualitatively, they are the minimal set which reproduces all salient features of the data after 20 fs delay. Additional parameters do not introduce improvements discernible by eye. (6) They are physically motivated by the known physics of the system as they relate exclusively to the $\pi^*$, $d_\parallel$ and Drude terms which are known to exhibit the largest changes through the phase transition.

## Time-dependent simulations of a dimer chain

These calculations are fully described in Zhang et al. [31]. We consider the $d_\parallel$ singlet and one $\pi^*$ orbital to simplify the theoretical model without losing the important physics of VO$_2$. Since the Peierls instability mainly occurs in the $c_R$ axis, we model VO$_2$ as a quasi-one-dimensional system, for which the displacement $\mathbf{X} \equiv (X_1, X_2)$ is introduced to capture the dimerizing displacement along $c_R$ axis and the band-splitting displacement in the perpendicular plane, respectively. The total Hamiltonian is given by

$$H = H_e + H_{e-\mathbf{X}} + \Phi(\mathbf{X}). \quad (2)$$

Here the pure electronic part reads

$$H_e = -\sum_i \sum_{a=1,2} \sum_{\sigma=\uparrow,\downarrow} t_a c_{a,\sigma,i}^\dagger c_{a,\sigma,i+1} - t_{12} \sum_i \sum_{\sigma=\uparrow,\downarrow} c_{1,\sigma,i}^\dagger c_{2,\sigma,i} + \text{H.c.}$$
$$+ \sum_i \sum_{a=1,2} \varepsilon_a n_{a,i} + \frac{U}{2} \sum_i n_i(n_i - 1), \quad (3)$$

where $a = 1, 2$ denotes the $d_\parallel$ and $\pi^*$ orbital, respectively, $t_a$ is the nearest-neighbour intra-orbital hopping, $t_{12}$ is the onsite inter-orbital hopping, and $\varepsilon_a$ is the onsite energy potential. We have

$$n_{a,i} = \sum_{\sigma=\uparrow,\downarrow} n_{a,\sigma,i}, \quad (4)$$

$$n_i = \sum_{a=1,2} \sum_{\sigma=\uparrow,\downarrow} n_{a,\sigma,i}, \quad$$

and $U$ is the onsite Hubbard repulsive interaction. The lattice distortion is modelled through the classical potential energy[3]

$$\Phi(X) = L\left[\frac{\alpha}{2}(X_1^2 + X_2^2) + \frac{\beta_1}{4}(2X_1X_2)^2 + \frac{\beta_2}{4}(X_1^2 - X_2^2)^2 + \frac{\gamma}{6}(X_1^2 + X_2^2)^3\right], \quad (5)$$

which is obtained from the Landau functional for improper ferroelectrics expanded up to the sixth order in the lattice displacements. Here $L$ is the number of lattice sites. For the electron-lattice coupling, we have

$$H_{e-\mathbf{X}} = -gX_1 \sum_i (-1)^i n_{1,i} - \frac{\delta}{2}X_2^2 \sum_i (n_{1,i} - n_{2,i}), \quad (6)$$

where the first term with coupling constant $g$ describes the dimerization induced by the displacement $X_1$ along $c_R$ axis while the second term with strength $\delta$ represents the crystal field splitting generated by $X_2$. Here we take the half-bandwidth as the unit of energy and set $t_1 = t_2 = 0.5$ eV. The inter-orbital hopping is chosen as $t_{12} = 0.1$ eV. We also assume the centre of gravity for these two bands to be the same and set $\varepsilon_1 = \varepsilon_2 = 0$. The Hubbard interaction is $U = 0.6$ eV, and the parameters for the lattice potential are set as $\alpha = 0.155$ eV/Å$^2$, $\beta_1 = 17.5$ eV/Å$^4$, $\beta_2 = 2\beta_1$, and $\gamma = 672.2$ eV/ Å$^6$. Finally, we choose the electron-lattice coupling strength as $g = 5.28$ eV/Å and $\delta = 20$ eV/Å$^2$. The total system is at quarter filling. Though most of these parameters have not been constrained experimentally, they are reasonable for this type of system and yield good agreement in their derived quantitites. For instance the equilibrium positions of the displacements are then found to be X$_1 \approx 0.205$ Å and X$_2 \approx 0.165$ Å, close to the experimental values and with appropriate ratio[31]. The transition temperature from the M1 phase to the R phase is found to a factor two of the true value[31], and the lattice natural frequency within a factor of two from previous DFT calculations[28].

**Simulation of the time evolution.** We use the tensor network methods to simulate the time evolution of the system induced by light pulse, which couples to the electronic degrees of freedom through the Peierls substitution $t_\alpha \to t_\alpha e^{\frac{i}{\hbar}A_{pump}(t)}$. Here $A_{pump}(t)$ is the vector potential for electric field $E_{pump}(t) = E_{0,pump} e^{-(t-t_{0,pump})^2/2\sigma_{pump}^2} \cos[\omega_{pump}(t - t_{0,pump})]$. We start from the equilibrium M1 phase at zero temperature with $X_1 \approx 0.205$ Å and $X_2 \approx 0.165$ Å, which minimizes the internal energy $\Phi_{eff} = \Phi(\mathbf{X}) + \langle H_{e-\mathbf{X}}\rangle + \langle H_e\rangle$. The time evolution of the system is decomposed into two parts, i.e., the electronic and lattice degrees of freedom. For the evolution of electronic state $|\psi\rangle$, we use the Born-Oppenheimer approximation within each time step $\delta t$, i.e., the lattice distortions are approximated as fixed, while the electronic degrees of freedom are dynamic. The corresponding equation of motion is given by the Schrödinger equation and can be written as

$$|\psi(t + \delta t)\rangle = e^{-iH[t, X(t)]\delta t/\hbar}|\psi(t)\rangle, \quad (7)$$

which can be simulated by the infinite time-evolving block decimation method. On the other hand, we use the classical approximation for the lattice dynamics and invoke the Ehrenfest theorem for the motion of lattice degrees of freedom

$$M\frac{d^2X_i}{dt^2} = F_i(t) - \xi\frac{dX_i}{dt}, \quad (8)$$

where $M$ is the effective mass of ions, which is set as 1082.41 eV·fs$^2$/Å$^2$ in this work, and $\xi = 13.16$ eV·fs/Å is the damping coefficient used to model the effects of disordering[11]. The forces $F_i$ are obtained through the Hellmann-Feynman theorem and explicitly read

$$F_1 = \frac{g}{2}\sum_{i=1,2}\cos(Qi)\langle\psi|n_{1,i}|\psi\rangle - \alpha X_1 - 2\beta_1 X_1 X_2^2 - \beta_2 X_1(X_1^2 - X_2^2)$$
$$- \gamma X_1(X_1^2 + X_2^2)^2 \quad (9)$$

and

$$F_2 = \frac{\delta}{2}X_2\sum_{i=1,2}\langle\psi|(n_{1,i} - n_{2,i})|\psi\rangle - \alpha X_2 - 2\beta_1 X_1^2 X_2 + \beta_2 X_2(X_1^2 - X_2^2)$$
$$- \gamma X_2(X_1^2 + X_2^2). \quad (10)$$

**Time-dependent optical conductivity**. Given the knowledge of the electronic wave function $|\psi(t)\rangle$ under the action of an external field $A(t)$, the temporal evolution of the current, defined as

$$\langle J(t)\rangle = \langle \psi(t)|J(t)|\psi(t)\rangle \qquad (11)$$

with

$$J(t) = \frac{\delta H(t)}{\delta A(t)} = -\mathrm{i}\sum_{a,\sigma,i} t_a \left[ e^{\mathrm{i}\frac{e}{\hbar}A(t)} c_{a,\sigma,i}^\dagger c_{a,\sigma,i+1} - \mathrm{H.c.} \right] \qquad (12)$$

can be readily obtained. To calculate the optical conductivity for a nonequilibrium system induced by the pump pulse, we employ the pump-probe based method proposed in Ref. 39, where the temporal evolution of the system is traced twice in order to identify the response of the system with respect to the later probe pulse. The procedures are as follows. First, the time-evolution process induced by the pump pulse $A_{\mathrm{pump}}(t)$ in the absence of probe pulse is evaluated, which describes the nonequilibrium development of the system, and we obtain the current $\langle J_{\mathrm{pump}}(t)\rangle$. Second, in addition to the pump pulse, we also introduce a narrow probe pulse $A_{\mathrm{probe}}(t)$ centered at time $t_*$, which leads to the current $\langle J_{\mathrm{total}}(t)\rangle$. The subtraction of $\langle J_{\mathrm{pump}}(t)\rangle$ from $\langle J_{\mathrm{total}}(t)\rangle$ produces the variation of the current due to the presence of probe pulse, i.e., $\langle J_{\mathrm{probe}}(t)\rangle$, with which the time-dependent optical conductivity at time $t_*$ can be calculated through

$$\sigma(\omega) = \frac{J_{\mathrm{probe}}(\omega)}{\mathrm{i}(\omega + \mathrm{i}\eta)LA_{\mathrm{probe}}(\omega)}, \qquad (13)$$

where $J_{\mathrm{probe}}(\omega)$ and $A_{\mathrm{probe}}(\omega)$ are the Fourier transformation of $\langle J_{\mathrm{probe}}(t)\rangle$ and $A_{\mathrm{probe}}(t)$, respectively. Numerically, a damping factor $e^{-\eta t}$ is introduced in the Fourier transformations, and the same $\eta$ is included in the denominator of the above equation.

### Density functional theory simulations

The DFT calculations were carried out using a plane-wave basis. We use the projector-augmented wave method[40], which is implemented in the Vienna Ab initio Simulation Package (VASP)[41]. The exchange and correlation were described using a tuned PBE0 hybrid functional[42,43] with 7% Hartree-Fock exchange, same as ref. 33. For the vanadium pseudopotential, 13 electrons ($3s^2 3p^6 3d^3 4s^2$) were treated as valence electrons. For the oxygen pseudopotential, six electrons ($2s^2 2p^4$) were treated as valence electrons. The cutoff energy of the plane-wave was 400 eV. For structural optimizations, we used a $\Gamma$-centered $3\times3\times3$ grid for the M1 and M0 phases (12 atoms per unit cell) and a $4\times4\times6$ grid for the R phase (six atoms per unit cell). The electronic self-consistent calculations were converged to $10^{-4}$ eV between successive iterations, and the structural relaxations were converged to $10^{-3}$ eV between two successive ionic steps. The densities of states were calculated using a $\Gamma$-centered $7\times9\times7$ grid for the M1 and M0 phase and a $\Gamma$-centered $9\times9\times15$ grid for the R phase. The crystal structures of the M1, M0 and R phases are shown in Supplementary Fig. 10.

### Data availability

Source data and plotting routines which reproduce all plots in Figs. 2 and 3 and Supplementary Figs. 1–5 and 7–9, as well as the atomic coordinates for our DFT calculations are available at the following Zenodo repository: https://doi.org/10.5281/zenodo.10610250.

### Code availability

All code necessary for processing the data is available at the following Zenodo repository: https://doi.org/10.5281/zenodo.10610250. The code for the tensor-network simulations and density-functional theory calculations is available from the authors upon request.

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

## Acknowledgements

This work was funded by the European Research Council (ERC) under the European Union's Horizon 2020 research and innovation programme: Starting Grant agreement HISOL no. 679649 and ERC Consolidator Grant XSOL no. 101001534. C.B. and J.C.T. acknowledge support from the United Kingdom's Engineering and Physical Sciences Research Council: Grant agreement EP/T020903/1. C.B. acknowledges support from the Royal Academy of Engineering through Research Fellowship No. RF/202122/21/133. This work was funded by the Spanish AIE (projects PID2022–137817NA-I00 and EUR2022–134052). A.S.J. acknowledges the support of the Ramón y Cajal Program (Grant RYC2021-032392-I). IMDEA Nanociencia acknowledges support from the "Severo Ochoa" Programme for Centers of Excellence in R&D (MICIN, CEX2020-001039-S). Computational resources were provided by the High-Performance Computing Center at the University of Memphis (X.S.). U.B. is also grateful for the financial support of the IBM Quantum Researcher Program. R.W.C. acknowledges support from the Polish National Science Centre (NCN) under the Maestro Grant No. DEC−2019/34/A/ST2/00081. T.G. acknowledges funding by Gipuzkoa Provincial Council (QUAN-000021-01), by the Department of Education of the Basque Government through the IKUR strategy and through the project PIBA_2023_1_0021 (TENINT), by the Agencia Estatal de Investigación (AEI) through Proyectos de Generación de Conocimiento PID2022-142308NA-I00 (EXQUSMI), by the BBVA Foundation (Beca Leonardo a Investigadores en Física 2023). The BBVA Foundation is not responsible for the opinions, comments and contents included in the project and/or the results derived therefrom, which are the total and absolute responsibility of the authors. S.T.P. acknowledges funding from the U. S. Department of Energy, Office of Science, Basic Energy Sciences, Materials Science and Engineering Directorate grant No. DE-FG02-09ER46554 and by the McMinn Endowment at Vanderbilt University. K.A.H. and R.F.H. acknowledge support from the U. S. National Science Foundation (EECS-1509740) and the Stevenson Endowment at Vanderbilt University. The ICFO group acknowledges support from: ERC AdG NOQIA; MCIN/AEI (PGC2018-0910.13039/501100011033, CEX2019-000910-S/10.13039/501100011033, Plan National FIDEUA PID2019-106901GB-I00, Plan National STAMEENA PID2022-139099NB-I00 project funded by MCIN/AEI/10.13039/501100011033 and by the "European Union NextGenerationEU/PRTR" (PRTR-C17.I1), FPI); QUAN-TERA MAQS PCI2019-111828–2); QUANTERA DYNAMITE PCI2022-132919 (QuantERA II Programme co-funded by European Union's Horizon 2020 program under Grant Agreement No 101017733), Ministry of Economic Affairs and Digital Transformation of the Spanish Government through the QUANTUM ENIA project call – Quantum Spain project, and by the European Union through the Recovery, Transformation, and Resilience Plan – NextGenerationEU within the framework of the Digital Spain 2026 Agenda; Fundació Cellex; Fundació Mir-Puig; Generalitat de Catalunya (European Social Fund FEDER and CERCA program, AGAUR Grant No. 2021 SGR 01452, QuantumCAT \ U16-011424, co-funded by ERDF Operational Program of Catalonia 2014-2020); Barcelona Supercomputing Center MareNostrum (FI-2023-1-0013); EU Quantum Flagship (PASQuanS2.1, 101113690); EU Horizon 2020 FET-OPEN OPTOlogic (Grant No 899794); EU Horizon Europe Program (Grant Agreement 101080086 — NeQST), ICFO Internal "QuantumGaudi" project; European Union's Horizon 2020 program under the Marie Sklodowska-Curie grant agreement No 847648; "La Caixa" Junior Leaders fellowships, "La Caixa" Foundation (ID 100010434): CF/BQ/PR23/11980043. Views and opinions expressed are, however, those of the author(s) only and do not necessarily reflect those of the European Union, European Commission, European Climate, Infrastructure and Environment Executive Agency (CINEA), or any other granting authority. Neither the European Union nor any granting authority can be held responsible for them. JO would like to thank Tom Tiwald and Inga Potsch of J.A. Woollam Co. for fruitful discussions.

## Author contributions

C.B., J.C.T., and A.S.J. conceived the study. R.H. and K.H. grew the $VO_2$ samples. C.B. and J.C.T. designed the ultra-broadband few-femtosecond spectroscopy experiments, and C.B. constructed the apparatus and performed the experiments. M.R., J.O., and A.S.J. performed additional characterization experiments. X.S. and S.T.P. developed the density functional theory model and X.S. did the calculations. L.Z., U.B., T.G., R.W.C., M.L., and A.S.J. developed the TN-MF model, while L.Z. did the calculations. C.B. and A.S.J. analysed the data and wrote the paper with contributions from all authors.

## Competing interests

The authors declare no competing interests.
