## [Transparent Peer Review file · Nature Communications]

Decoupled few-femtosecond phase transitions in vanadium dioxide

Corresponding Author: Dr Christian Brahms

Version 0:

Reviewer comments:

Reviewer #1

(Remarks to the Author)

The authors did a good job responds to my previous questions. I can recommend it for publication in nature communications.

Reviewer #3

(Remarks to the Author)

Brahms, et al. have made meaningful revisions which may make their work appropriate for Nature Communications.

In my view, there are two outstanding issues:

(1) To reflect the fixed-frequency nature of the photoexcitation process, film specifics, etc. the authors should change the wording of the sentence

‘Our results resolve the complete structural and electronic nature of the light-induced phase transition in VO₂ and establish ultrabroadband few-femtosecond spectroscopy as a powerful new tool for studying quantum materials out of equilibrium.’

to one of the following or a suitable substitution.

(a) ‘Our results resolve the structural and electronic nature...’

(b) ‘Our results resolve the complete structural and electronic nature of a light-induced phase transition...’

(2) The authors state that the length scale is somewhat arbitrary in their parameterization of the lattice modes because of their dimensionless parameterization of the lattice potential energy and electron-phonon interactions.

Transparency, reproducibility, internal consistency, and confidence in as-drawn conclusions would require parameters which are not only qualitatively consistent with other experimental features (e.g. a hierarchy of scales) but are at least in the ballpark of something sensible. While the authors have shown in their rebuttal and revised manuscript that the ratio of X1 and X2 is consistent with experiment, it is unclear to a reader that the scales of the electronic, lattice, and electron-phonon terms are appropriate. For example, t should be related to the bandwidth; $L \cdot \alpha$ should be connected to the natural frequency of X1 and X2, given the reduced mass of these coordinates; and similar statements for the beta and gamma terms and the adjusted frequency. The latter are central to the argument of fast phonon dynamics discussed by the authors and require the length scale to be similarly fixed for consistency. I suggest the authors address this connection in an appendix, to minimize alterations of the current manuscript body.

In my view, it is unclear how a reader can give credence to the conclusions drawn from the theory in the absence of this connection.

Reviewer #4

(Remarks to the Author)

It is very much appreciated that the use of broadband optical measurements disambiguates effects of charge screening/photoexcited carriers with an actual photoinduced phase transition by seeing changes to transitions explicitly, something which is lacking from many previous studies on photoinduced insulator-to-metal transitions.

The authors do give an explanation of the fitting procedure in the methods section but it is still a bit unclear how robust the extraction of the extremely short drude scattering time is to this fitting procedure. There aren't any error bars given or figures showing the failure of being able to fit the data with a more conventional, longer scattering time, such as that seen in the thermal state of VO₂. The novelty of what is found in the early time measurements shown in the manuscript, which do give unprecedented temporal resolution to the early-time dynamics of the photoinduced phase transition, does seem to rely on measuring such a short drude scattering time. There are a few reasons why the fitting of the drude parameters may be of particular concern, and more information is needed. The first concern is that it appears, based on figure 2j, that the plasma frequency and drude damping are pretty close to each other in size, so they can have somewhat similar contributions to the strength of the drude term. Looking at Supplementary figure S3 (which only shows magnitude of reflectivity change), it's also not particularly clear to me that the changes to the d|| peak do not give reflectivity changes at some energies that are also sensitive to the drude parameters, given the large plasma frequency and drude damping that are extracted. Looking at raw reflectivity of the films on the same substrate from Fu et al 2013 (<https://doi.org/10.1063/1.4788804>) there are certainly features in the reflectivity other than the change to the drude component that appear in the energy range considered. Some more information or supplementary figures showing the robustness of the fit and extraction of the unprecedented short scattering time in the drude component are necessary since some of the novelty of the interpretation of the data, such as the pump-time evolution of the conductivity relies on it being much longer than this extracted drude scattering time. Specifically the conclusion that 100fs is in this case an anomalously long time constant relies on the relationship to the drude scattering time.

The authors also state that there are no O₂p holes in the system because the pump energy is not large enough. The fluence used is pretty high, not anomalously high for driving the transition in VO₂, but high. I would imagine at these high fluences multi-photon absorption can occur as well, is there a way to confirm this is not a significant contribution to the results.

Reviewer #5

(Remarks to the Author)

Version 1:

Reviewer comments:

Reviewer #3

(Remarks to the Author)

The authors have addressed all of my concerns. I recommend publication.

Reviewer #4

(Remarks to the Author)

The authors have done a sufficient job to respond to the points in my review of their previous manuscript draft. I do recommend this work for publication in nature communications.

Reviewer #5

(Remarks to the Author)

Reviewer #1 (Remarks to the Author):

The authors did a good job responds to my previous questions. I can recommend it for publication in nature communications.

We thank the reviewer for their careful reading and their recommendation to publish our work in Nature Communications.

Reviewer #3 (Remarks to the Author):

Brahms, et al. have made meaningful revisions which may make their work appropriate for Nature Communications.

We thank the reviewer for recognizing the substantial nature of the revisions made thusfar, and welcome the opportunity to address their remaining concerns.

In my view, there are two outstanding issues:

(1) To reflect the fixed-frequency nature of the photoexcitation process, film specifics, etc. the authors should change the wording of the sentence
'Our results resolve the complete structural and electronic nature of the light-induced phase transition in VO₂ and establish ultrabroadband few-femtosecond spectroscopy as a powerful new tool for studying quantum materials out of equilibrium.'

to one of the following or a suitable substitution.

(a) 'Our results resolve the structural and electronic nature...'

(b) 'Our results resolve the complete structural and electronic nature of a light-induced phase transition...'

We have modified the text to 'Our results resolve the structural and electronic nature...' as suggested by the reviewer. We agree that some aspects of the process may indeed be modified by other parameters not explored in our study.

(2) The authors state that the length scale is somewhat arbitrary in their parameterization of the lattice modes because of their dimensionless parameterization of the lattice potential energy and electron-phonon interactions.

Transparency, reproducibility, internal consistency, and confidence in as-drawn conclusions would require parameters which are not only qualitatively consistent with other experimental features (e.g. a hierarchy of scales) but are at least in the ballpark of something sensible. While the authors have shown in their rebuttal and revised manuscript that the ratio of X1 and X2 is consistent with experiment, it is unclear to a reader that the scales of the electronic, lattice, and electron-phonon terms are appropriate. For example, t should be related to the bandwidth; $L^*\alpha$ should be connected to the natural frequency of X1 and X2, given the reduced mass of these coordinates; and similar statements for the beta and gamma terms and the adjusted frequency. The latter are

central to the argument of fast phonon dynamics discussed by the authors and require the length scale to be similarly fixed for consistency. I suggest the authors address this connection in an appendix, to minimize alterations of the current manuscript body.

We have now addressed the reviewers concerns by fixing a length scale of $\ell \approx 0.1 \text{ \AA}$ along with the half-bandwidth of 1 eV, which leads to $X_1 \approx 0.205 \text{ \AA}$ and $X_2 \approx 0.165 \text{ \AA}$, which are close to real lattice parameters of VO₂. In this length scale, the electron-lattice couplings and lattice potential parameters λ (i.e., α , $\lambda_{1,2}$, γ , g , and δ) that couple to $X_{1,2}^{\nu}$ transform to λ/ℓ^{ν} . This leads to the following parameters:

$t_1 = t_2 = 0.5 \text{ eV}$, $t_{12} = 0.1 \text{ eV}$, $\epsilon_1 = \epsilon_2 = 0$, $U = 0.6 \text{ eV}$, $\alpha = 15.5 \text{ eV/\AA}^2$, $\lambda_1 = 17.5 \text{ eV/\AA}^4$, $\lambda_2 = 2\lambda_1$, $\gamma = 672.2 \text{ eV/\AA}^6$, $g = 5.28 \text{ eV/\AA}$, $\delta = 20 \text{ eV/\AA}^2$

Most of these parameters are not known from experiment for VO₂ and cannot be compared to check their realism beyond simple order of magnitude comparisons to other materials, which is why we had not previously included such an assignment. However, we can compare some derived quantities to show they provide a realistic description of VO₂, as detailed in Reference 30. These include the lattice natural frequency (46 fs, factor of two from previous DFT results in PRB 88, 035119 (2013)), transition temperature (734K, $\approx 2x$ from experiment), and optical bandgap (1 eV, within factor of two from experimental 0.8 eV). We have modified the text in the methods section as follows:

Here we take the half-bandwidth as the unit of energy and set $t_1 = t_2 = 0.5 \text{ eV}$. The inter-orbital hopping is chosen as $t_{12} = 0.1 \text{ eV}$. We also assume the centre of gravity for these two bands to be the same and set $\epsilon_1 = \epsilon_2 = 0$. The Hubbard interaction is $U = 0.6 \text{ eV}$, and the parameters for the lattice potential are set as $\alpha = 0.155 \text{ eV/\AA}^2$, $\lambda_1 = 17.5 \text{ eV/\AA}^4$, $\lambda_2 = 2\lambda_1$, and $\gamma = 672.2 \text{ eV/\AA}^6$. Finally, we choose the electron-lattice coupling strength as $g = 5.28 \text{ eV/\AA}$ and $S = 20 \text{ eV/\AA}^2$. The total system is at quarter filling. Though most of these parameters have not been constrained experimentally, they are reasonable for this type of system and yield good agreement in their derived quantities. For instance the equilibrium positions of the displacements are then found to be $X_1 \approx 0.205 \text{ \AA}$ and $X_2 \approx 0.165 \text{ \AA}$, close to the experimental values and with appropriate ratio³¹. The transition temperature from the M1 phase to the R phase is also found to a factor two of the true value³¹, and the lattice natural frequency within a factor of two from previous DFT calculations³⁹.

And included the units elsewhere as appropriate and adjusted the axis labels in Figure 3.

In my view, it is unclear how a reader can give credence to the conclusions drawn from the theory in the absence of this connection.

We hope our illustration that our model successfully recreates a wide range of experimentally determinable parameters with reasonable assumptions for the various coupling constants is sufficient to convince the reviewer that the model can indeed be used to explain new behaviours of VO₂.

Reviewer #4 (Remarks to the Author):

It is very much appreciated that the use of broadband optical measurements disambiguates

effects of charge screening/photoexcited carriers with an actual photoinduced phase transition by seeing changes to transitions explicitly, something which is lacking from many previous studies on photoinduced insulator-to-metal transitions.

We thank the reviewer for noting this important distinction between our work and previous studies on photoinduced phase transitions.

The authors do give an explanation of the fitting procedure in the methods section but it is still a bit unclear how robust the extraction of the extremely short drude scattering time is to this fitting procedure. There aren't any error bars given or figures showing the failure of being able to fit the data with a more conventional, longer scattering time, such as that seen in the thermal state of VO₂. The novelty of what is found in the early time measurements shown in the manuscript, which do give unprecedented temporal resolution to the early-time dynamics of the photoinduced phase transition, does seem to rely on measuring such a short drude scattering time. There are a few reasons why the fitting of the drude parameters may be of particular concern, and more information is needed. The first concern is that it appears, based on figure 2j, that the plasma frequency and drude damping are pretty close to each other in size, so they can have somewhat similar contributions to the strength of the drude term. Looking at Supplementary figure S3 (which only shows magnitude of reflectivity change), it's also not particularly clear to me that the changes to the $d||$ peak do not give reflectivity changes at some energies that are also sensitive to the drude parameters, given the large plasma frequency and drude damping that are extracted. Looking at raw reflectivity of the films on the same substrate from Fu et al 2013 (<https://doi.org/10.1063/1.4788804>) there are certainly features in the reflectivity other than the change to the drude component that appear in the energy range considered. Some more information or supplementary figures showing the robustness of the fit and extraction of the unprecedented short scattering time in the drude component are necessary since some of the novelty of the interpretation of the data, such as the pump-time evolution of the conductivity relies on it being much longer than this extracted drude scattering time. Specifically the conclusion that 100fs is in this case an anomalously long time constant relies on the relationship to the drude scattering time.

This is an important point, and one we are happy to address. The reviewer is correct that across the range of interest we have multiple contribution to the reflectivity, both from the $d||$ peak and the Drude terms, and furthermore that both the Drude plasma frequency and damping contribute to the strength of the Drude term. This is why the broadband nature of our experiment is absolutely essential; while these terms all affect the reflectivity, they do so with very different spectral dependencies. In particular the Drude plasma can be viewed as a step-like function of frequency in the dielectric function, where the plasma term changes the frequency at which the step occurs, while the damping term affects the sharpness of this step. The spectral dependence of the Drude term thus encodes both of this terms uniquely. In our case the presence of the $d||$ term complicates this assignment, but our bandwidth is sufficient that the found solution is quite unique. To illustrate this, we have added extended data Fig. S4 (also reproduced below), which shows the fitting result in which the Drude damping term is capped at 1.35 eV (slightly above the equilibrium value of 0.9 eV). All parameters of the Gaussian $d||$ around 2 eV are allowed to vary; as can be seen the quality of the fit decreases dramatically. This results because the Drude plasma frequency alone is *not*

sufficient to compensate the contribution from the Drude damping across a large bandwidth and the Gaussian term must change significantly to fit the lower photon energy region, which in turn

Extended Data Fig. S4: Fit with clamped damping. **a** Experimental data (same as Fig. 2a). **b** Full fit with free parameters as discussed in the main text (same as Fig. 2c). **c** Result of fitting with the Drude damping term clamped to 1.35 eV (slightly above the value in the rutile phase through heating or at long time delays). The fit with restricted damping is clearly worse, indicating that the response cannot be explained without the damping contribution.

degrades the fitting at higher photon energies.

We have additionally replotted the fit parameters with error bars derived from the covariance matrix of the fit in Fig. 2, reproduced here for convenience:

The standard error on each parameter is shown as the shaded region around each fit line. As can be seen the errors are very small, and even at the extrema support our conclusions regarding unprecedentedly high scattering.

The authors also state that there are no O2p holes in the system because the pump energy is not large enough. The fluence used is pretty high, not anomalously high for driving the transition $n \rightarrow VO_2$, but high. I would imagine at these high fluences multi-photon absorption can occur as well, is there a way to confirm this is not a significant contribution to the results.

We were originally referring to the photon energy when claiming the energy was not sufficient and not considering multiphoton effects, but we now address this possibility as well. The two-photon absorption cross section of VO_2 in the insulating monoclinic phase is known to be extremely small; previous z-scan measurements attempting to characterize the nonlinear absorption have been unable to observe any contribution in the M-phase (Appl. Phys. Lett. 85, 5191–5193 (2004) and Applied Surface Science, Volume 258, 5319–5322 (2012)). Only once the fluence was sufficient to trigger the phase transition via single-photon absorption were any nonlinear effects observed. From this we can conclude the nonlinear absorption in the M-phase has a cross section of less than $1 \text{ cm}^2/\text{GW}$ (Appl. Phys. Lett. 85, 5191–5193 (2004)). In our experiments the peak intensity inside the material is on the order of 10^{12} W/cm^2 , this gives an **upper bound** on the nonlinear absorption of 1000 cm^{-1} , two orders of magnitude smaller than the linear absorption length of 161290 cm^{-1} (Physical Review X, 10, 031047 (2020)). As such we can conclude that there is $<1\%$ contribution from nonlinear absorption including all orbitals, not solely those from the O2p states. The two-photon absorption from p to d orbitals should be further suppressed by symmetry arguments. We note

again this value is limited by the resolution of previous two-photon absorption measurements, as that two-photon absorption in the M-phase has never been experimentally detected, and an examination of the fluence dependence in our experiment (Fig. S2b) shows similar dependence as experiments with much longer pulses where multi-photon effects should be even less probable. As such we conclude multiphoton excitation has a negligible contribution to our experiments. In the text we now note:

“Multiphoton absorption is known to be extremely weak in the M1 phase²⁶ and the photon energy of our pump pulse is insufficient to directly excite holes in the $O''\#$ orbitals. The $O''\#$ orbitals themselves exhibit only marginal changes during the IMT²³. Therefore, the $O''\# \rightarrow \uparrow\uparrow$ and $O''\# \rightarrow *$ transitions effectively track the motion of the unoccupied portion of the $\uparrow\uparrow$ and $*$ bands above the Fermi level.”

Reviewer #5 (Remarks to the Author):

We thank the review for their contributions and their time.